# Meta-FAVAE: Toward Fast and Diverse Few-shot Image Generation via Meta-Learning and Feedback Augmented Adversarial VAE

**Fangli Ying**\*, **Aniwat Phaphuangwittayakul**\*, **Yi Guo, Xiaoyue Huang, Yue Wang**
East China University of Science and Technology, China
`yfangli@ecust.edu.cn, aniwat.pha@gmail.com`

## Abstract

Learning to synthesis realistic images of new categories based on just one or a few examples is a challenge task for deep generative models, which usually require to train with a large amount of data. In this work, we propose a data efficient meta-learning framework for fast adapting to few-shot image generation task with an adversarial variational auto-encoder and feedback augmentation strategy. By training the model as a meta-learner, our method can adapt faster to the new task with significant reduction of model parameters. We designed a novel feedback augmented adversarial variational auto-encoder. This model learns to synthesize new samples for an unseen category just by seeing few examples from it and the generated interpolated samples are then used in feedback loop to expand the inputs for encoder to train the model, which can effectively increase the diversity of decoder output and prevent the model collapse. Additionally, this method can also generalize to adapt to more complex color image generation tasks.

## 1 Introduction

Following the great success of deep learning, deep generative models, such as Generative Adversarial Network (GAN)( Goodfellow et al. (2014)) and Variational Autoencoder (VAE)( Kingma & Welling (2014)), achieved tremendous progress in the task of generating realistic image by training with large-scale datasets. However, due to the high expense of collecting and annotating a large amount of data, given only one or a few images from a category, it is still a challenging task for the typical deep generative models to learn fast and adapt to synthesis realistic and diverse images. This task is referred to as few-shot image generation(FSIG). Several recent efforts have been devoted to improve the learning efficiency of models for the FSIG task. These approaches can roughly fallen into three categories, that are transformation-based methods, fusion-based methods and optimization-based methods. For the transformation-based methods, DAGAN( Antoniou et al. (2018)) is an GAN-based approach that captures cross-class transformations on one conditional image and injects random vectors into the generator for generating different images. However, the diversity of generated images from this method is quite limited and it fails to capture the information from multiple images in the same category. Among the fusion-based methods, GMN( Bartunov & Vetrov (2018)) is inspired by the matching network( Vinyals et al. (2016)) and combines a matching procedure with VAE. Due to the limited generation capacity of GMN, this method can only generate binary digits. For generating more diversified and realistic images, MatchingGAN( Hong et al. (2020)) combines adversarial learning and matching generators and is capable to fully exploit several conditional images from the same class. Nevertheless, this method still struggles with more complex natural color image generation and training these models is time-consuming due to complexity of the networks. Optimization-based methods are proposed based on the concept of meta-learning and initialize a generator with images from seen categories and fine-tuning the trained model with images from each unseen category. Clouâtre & Demers (2019) proposed a FIGR by applying meta-learning algorithm Reptile( Nichol et al. (2018)) to GAN. It can significantly reduce the trainable parameters of the model and fast adapt to the new generation task in the few-shot setting. DAWSON Liang et al. (2020) is another method incorporating meta-learning model of MAML( Finn

---

\*Equal Contribution

et al. (2017)) with GAN for the few-shot generation task. Nevertheless, the DAWSON generates images without diversity and is inapplicable for color images. Both models adopted a similar idea of meta-learning method, but can hardly produce sharp and realistic images or adapt to complex natural color images. FAML ( Phaphuangwittayakul et al. (2021)) is another meta-learning inspired method that can significantly reduce the model parameters and fast adapt to generate images from color image dataset. This simple and effective model increased the number of generator iterations and utilized conditional feature vectors. However, given the insufficient input of unseen category, only conditioning the feature vectors from the encoder can not guarantee the exploitation of the key input features. Moreover, training GAN-based model is time-consuming and mode collapse issues are remained, while VAEs have limited capacity of generating high-quality images.

To tackle the above problems, we propose a Meta-learning-based Feedback augmented Adversarial Variational Auto-Encoder (Meta-FAVAE),that learns optimal model parameters to fast adapt to few-shot image generation task and increases the diversity by a feedback augmentation strategy. This model is trained in a meta-learning mechanism with an inner-loop and outer-loop. By integrating meta-learning, this method has less than 10M parameters and thus leads to a faster adaption to the new generation tasks of unseen categories. In order to combine the advances from VAE and GAN ( Plumerault et al. (2021)), we introduced an optimal adversarial variational auto-encoder to generate high-quality images while retrieving key latent features. More specifically, the model utilized Relativistic average GAN ( Jolicoeur-Martineau (2018)) and beta-VAE ( Higgins et al. (2016)) to significantly improve data quality and stability of the generated images with $64 \times 64$ size outputs. By introducing a novel feedback augmentation technique, we can significantly increase the diversity of the generated images by adding interpolated samples to the inputs, which can also prevent model collapse. Additionally, we utilize the dual fusion of latent code and random noise vectors and mode-seeking regularization term from FAML for improving the generalization of few-shot image generation task to natural color images. Extensive experiments validate the superiority of our Meta-FAVAE in terms of fast adaptation and diverse image generation over other state-of-the-art methods.

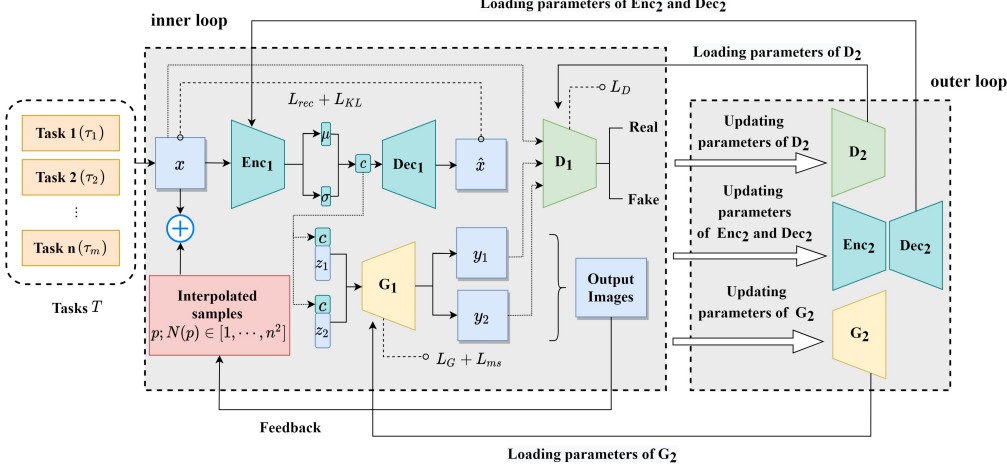

Figure 1: The framework follows a meta-learning mechanism. The inner loop contains an Adversarial-VAE including an encoder $Enc_1$, a decoder $Dec_1$, a generator $G_1$ and a discriminator $D_1$. The outer loop also has the same components containing an encoder $Enc_2$, a decoder $Dec_2$, a generator $G_2$ and a discriminator $D_2$. The feedback augmentation takes the generated images from generator $G_1$ in the inner loop and perform a feedback loop operation to add the interpolated samples into the inputs of $Enc_1$.

## 2 METHODOLOGY

Figure 1 illustrates the pipeline of proposed model in this study. Given a set of tasks $\{T_i\}_{i=1}^m$ containing various tasks where each task $\tau$ is an image generation for one class of images $(X_\tau)$. The

model can generate a set of images for the unseen classes. This framework is composed of two main phase, which are training phase and generation phase. During the inner loop of the training phase, the encoder $Enc_1$ of VAE extracts mean $\mu$ and variance $\sigma$ from images $x$ of the training task to form a latent code $c$. Then, the decoder $Dec_1$ of VAE reproduces images $\hat{x}$ from the latent code. The generator $G_1$ takes the latent code concatenated with two random noise vectors ($z_1$ and $z_2$) as the input and generates output images, $y_1 = G_1(z_1, c)$ and $y_2 = G_1(z_2, c)$. The discriminator network $D_1$ distinguishes the real images (input $x$) and fake images (output samples $y$ from generator network $G_1$). The Adam optimizer Kingma & Ba (2014) is used to optimize the encoder $Enc_1$, decoder $Dec_1$, generator $G_1$, and discriminator $D_1$ networks in the loss computation. The objective functions to calculate the loss of $Enc_1$, $Dec_1$, $G_1$, and $D_1$ are defined as follows.

Instead of training the model with only input images as traditional method, we increase the number of training images by fusing the interpolated samples. The interpolated samples are produced based on the images generated by our model. The linear interpolation is performed based on the four generated images in each episode. The linear interpolation is a technique to visualize the data in a subspace. With two images multiplied by the interpolation coefficients ranging in [0.1,0.9], the new examples between these two images can be explored in the space of generated images. The total number of interpolated samples are equal to loop $n^2$, where $n > 0$. For example, a total of 16 interpolated images with $n = 4$ are produced by random images generated by the latent code of Meta-AVAE and four random noise vectors. Once the interpolated images are generated, these images are used to combine with the input images and step the model with the inner loop gradient one more episode.

To train VAE for extracting the latent code and producing reconstructed images, a reconstruction loss and a Kullback-Leibler (KL) divergence Larsen et al. (2016) loss are computed as equation (1).

$$\mathcal{L}_{VAE} = \frac{1}{g} \sum (x - \hat{x})^2 + \beta \text{KL}(q(c|x) \parallel p(c)) \tag{1}$$

where $g$ denotes the number of images sampled from a class of images $X_\tau$. $\beta$ indicates adjustable hyperparameter, set to 10. The loss functions of both discriminator and generator followed the two random noise vectors $z_1$ and $z_2$ as follows.
The Discriminator objective function is:

$$\mathcal{L}_{disc,1} = -\mathbb{E}_x[\log(D_1(x)] - \mathbb{E}_{y_1}[\log(1 - D_1(y_1)] \tag{2}$$

$$\mathcal{L}_{disc,2} = -\mathbb{E}_x[\log(D_1(x)] - \mathbb{E}_{y_2}[\log(1 - D_1(y_2)] \tag{3}$$

Thus, $\mathcal{L}_{disc}$ is the average of $\mathcal{L}_{disc,1}$ and $\mathcal{L}_{disc,2}$ and the Generator objective function is as below and $\mathcal{L}_{gen}$ is the average of $\mathcal{L}_{gen,1}$ and $\mathcal{L}_{gen,2}$ :

$$\mathcal{L}_{gen,1} = -\mathbb{E}_{y_1}[\log(D_1(y_1)] - \mathbb{E}_x[\log(1 - D_1(x)] \tag{4}$$

$$\mathcal{L}_{gen,2} = -\mathbb{E}_{y_2}[\log(D_1(y_2)] - \mathbb{E}_x[\log(1 - D_1(x)] \tag{5}$$

For each sampled task $\tau$ including training and test tasks, Meta-FAVAE first initializes and then minimizes the parameters of the encoder $\Phi_{1,\text{enc}}$, decoder $\Phi_{1,\text{dec}}$, discriminator $\Phi_{1,\text{disc}}$, and generator $\Phi_{1,\text{gen}}$ in the $K$ inner loop through objective functions $\mathcal{L}_{VAE}$, $\mathcal{L}_{disc}$, and $\mathcal{L}_{gen}$, respectively. The local minimum parameters of the encoder, decoder, discriminator, and generator are achieved by minimizing loss $\mathcal{L}_{VAE}$, $\mathcal{L}_{disc}$, and $\mathcal{L}_{gen}$. The total loss function of our Meta-FAVAE method can be written as

$$\mathcal{L} = \mathcal{L}_{VAE} + \mathcal{L}_{disc} + \mathcal{L}_{gen} + \lambda_{ms}\mathcal{L}_{ms} \tag{6}$$

Where $\mathcal{L}_{ms}$ represents a mode-seeking regularization termMao et al. (2019) that maximizes the ratio of the distance between $G_1(z_1, c)$ and $G_1(z_2, c)$ corresponding to the distance between $z_1$ and $z_2$, as expressed in equation (7).

$$\mathcal{L}_{ms} = \max_{G_1}\left(\frac{d(G_1(z_1, c), G_1(z_2, c))}{d(z_1, z_2)}\right) \tag{7}$$

where $d(\cdot, \cdot)$ indicates the $L_1$ norm distance metric. Following the same settingMao et al. (2019), we set the hyper-parameters $\lambda_{ms} = 1$. The outer loop is performed after the $K$ steps of the inner loop operation. The global parameters $\Phi_2$ of the encoder $Enc_2$, decoder $Dec_2$, generator $G_2$, and discriminator $D_2$ are updated with parameters $\Phi_1$ by setting it to $\Phi_2 - \Phi_1$. The pseudo-code of Meta-FAVAE for the training process and generation process is described in supplementary

## 3 EVALUATIONS AND CONCLUSION

The experiments are conducted on three few-shot image datasets: MNIST( LeCun & Cortes (2010)), Omniglot( Lake et al. (2011)) and VGG-Faces( Cao et al. (2018)).The quality of generated images are evaluated by FID ( Heusel et al. (2017)), while the diversity of output are evaluated through LPIPS( Zhang et al. (2018)) and IS( Xu et al. (2018)).Based on the 1000 generated images of the unseen classes, we compare our model with baseline approaches including DAGAN( Antoniou et al. (2018)), FIGR( Clouâtre & Demers (2019)), DAWSON( Liang et al. (2020)) and FAML( Phaphuangwittayakul et al. (2021)). The results presented in Table 1 show our method achieves the best performance in terms of quality and diversity in the evaluations on all three datasets. Additionally, the feedback operation is leveraged to increase the diversity of output images and generalize the model across different categories. The model with feedback operation not only augments the data to increase the size of input but also avoids the model collapse by only a few training samples. Although a variational auto-encoder has been added to the network, our model can significantly reduce the number of parameters to 8.3M, which is merely one-quarter of FIGR. Furthermore, it converges over ten times faster than the baseline models. Thus, our model can fast adapt to new generation tasks with limited steps as shown in Figure 2. Comparing with the generated images from other methods, the Meta-FAVAE can not only retain the compact features due to the latent representations but also generate a wide variety of samples within limited episodes by using feedback augmentation.

In this work, we have introduced a novel FSIG approach based on meta-learning with a feedback augmented variational auto-encoder. This model can outperform the existing approaches in terms of fast adaptation and the diversity of outputs.

Table 1: The evaluation metrics for the images of unseen class generated by baseline and our methods on three datasets.

| Method | MNIST FID ($\downarrow$) | Omniglot FID ($\downarrow$) | VGG-Faces FID ($\downarrow$) | IS ($\uparrow$) | LPIPS ($\downarrow$) |
|---|---|---|---|---|---|
| DAGAN | 78.56 | 72.23 | 196.12 | 2.22 | 0.2674 |
| FIGR | 36.62 | 68.95 | 250.14 | 2.85 | 0.6701 |
| DAWSON | 86.72 | 75.59 | 184.15 | 2.56 | 0.8672 |
| FAML | 33.94 | 78.69 | 42.72 | 7.10 | 0.4011 |
| Meta-AVAE | 28.65 | **63.29** | 54.44 | 11.53 | 0.2431 |
| Meta-FAVAE | **24.23** | 73.74 | **42.11** | **15.58** | **0.2056** |

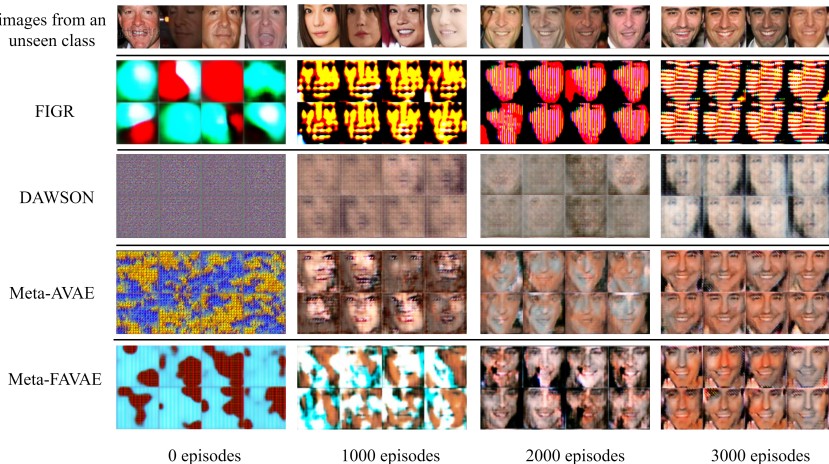

Figure 2: The generated images from the VGG-Face dataset by the baseline meta-learning based models (FIGR, DAWSON) and our Meta-FAVAE model.

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

## A  META-AVAE TRAINING AND GENERATION PROCESSES

The pseudo-code of Meta-AVAE training algorithm is described in in Algorithm 1.

Besides, the generation process is performed using the gradient from training process to generate the images from unseen class. The Meta-AVAE generation process is described in Algorithm 2.

## B  ADDITIONAL COMPARISONS

We compare the generated output images between FAML and Meta-FAVAE. The samples from MNIST, Omniglot, and VGG-Face datasets are selected as representative for representing the performance of generative models. It can be observed that Meta-FAVAE with loopback operation can generate more diverse samples than FAML as shown in Figure 3

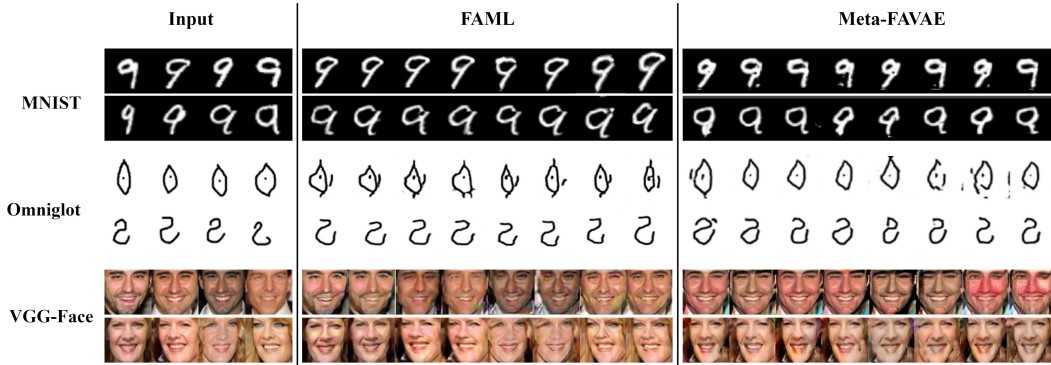

Figure 3: The comparison of generated output images generated by FAML and Meta-FAVAE on MNIST, Omniglot, and VGG-Face datasets with sample images.

## C  MORE GENERATED RESULTS

The samples of multiple generated output images that are produced by the Meta-FAVAE on MNIST, Omniglot, and VGG-Face datasets are illustrated in Figure 4, 5, 6, respectively. The images in blue

---

**Algorithm 1** Meta-AVAE training process The parameters of the implemented model in the paper are defaulted as: $n_{iter} = 100000$, $n_{VAE} = 5$, $n_D = 1$, $n_G = 5$, $g = 4$, $K = 10$, $\alpha_{1,\text{enc}} = \alpha_{1,\text{dec}} = 0.0001$, $\alpha_{1,\text{disc}} = \alpha_{1,\text{gen}} = 0.001$, $\alpha_{2,\text{enc}} = \alpha_{2,\text{dec}} = 0.00001$, $\alpha_{2,\text{disc}} = \alpha_{2,\text{gen}} = 0.0001$

---

**Require**: $\Phi_1$, parameters of $Enc_1$, $Dec_1$, $G_1$, and $D_1$. $\Phi_2$, parameters of $Enc_2$, $Dec_2$, $G_2$, and $D_2$. $n_{iter}$, number of global iterations called episodes. $n_{VAE}$, number of variational autoencoder iterations. $n_D$, number of discriminator iterations. $n_G$, number of generator iterations. $g$, batch size of images, $K$, number of inner loop iterations. $\alpha$'s, learning rate.

1: Initialize $\Phi_2$
2: **for** $i < n_{iter}$ **do**
3:     Set $\Phi_1 = \Phi_2$
4:     Sample training task $\tau$
5:     Sample $g$ images as $x$ from $X_\tau$
6:     **for** $i < K$ **do**
7:         # Train VAE
8:         **for** $j < n_{VAE}$ **do**
9:             $c, \mu, \sigma \leftarrow Enc_1(x)$
10:            $\hat{x} \leftarrow Dec_1(c)$
11:            $\Phi_{1,\text{enc}} \leftarrow \Phi_{1,\text{enc}} + \alpha_{1,\text{enc}} \nabla_{\Phi_1} \{ \mathcal{L}_{rec} + \mathcal{L}_{KL} \}$
12:            $\Phi_{1,\text{dec}} \leftarrow \Phi_{1,\text{dec}} + \alpha_{1,\text{dec}} \nabla_{\Phi_1} \{ \mathcal{L}_{rec} + \mathcal{L}_{KL} \}$
13:         **end for**
14:         # Train discriminator $D_1$
15:         **for** $j < n_D$ **do**
16:            $z_1 \leftarrow \mathcal{N}(0,1)$, $z_2 \leftarrow \mathcal{N}(0,1)$
17:            $y_1 \leftarrow G_1(z_1, c)$, $y_2 \leftarrow G_1(z_2, c)$
18:            $\Phi_{1,\text{disc}} \leftarrow \Phi_{1,\text{disc}} + \alpha_{1,\text{disc}} \nabla_{\Phi_1} \{ \mathcal{L}_{disc} \}$
19:         **end for**
20:         # Train generator $G_1$
21:         **for** $j < n_G$ **do**
22:            $z_1 \leftarrow \mathcal{N}(0,1)$, $z_2 \leftarrow \mathcal{N}(0,1)$
23:            $y_1 \leftarrow G_1(z_1, c)$, $y_2 \leftarrow G_1(z_2, c)$
24:            $\Phi_{1,\text{gen}} \leftarrow \Phi_{1,\text{gen}} + \alpha_{1,\text{gen}} \nabla_{\Phi_1} \{ \mathcal{L}_{gen} + \mathcal{L}_{ms} \}$
25:         **end for**
26:     **end for**
27:     Set gradient $\Phi_2$ of $Enc_2, Dec_2, D_2, G_2$ to $\Phi_2 - \Phi_1$
28:     Perform step of Adam update on $\Phi_2$ with learning rate $\alpha_{2,\text{enc}}, \alpha_{2,\text{dec}}, \alpha_{2,\text{disc}}, \alpha_{2,\text{gen}}$
29: **end for**

---

---

**Algorithm 2** Meta-AVAE generation process

---

1: Using $\Phi_2$ from the training process
2: Set $\Phi_1 = \Phi_2$
3: Sample test task $\tau$
4: Sample $g$ images as $x$ from $X_\tau$
5: **for** $i < K$ **do**
6:    # Train VAE
7:    **for** $j < n_{VAE}$ **do**
8:      $c, \mu, \sigma \leftarrow Enc_1(x)$
9:      $\hat{x} \leftarrow Dec_1(c)$
10:     $\Phi_{1,\text{enc}} \leftarrow \Phi_{1,\text{enc}} + \alpha_{1,\text{enc}} \nabla_{\Phi_1} \{ \mathcal{L}_{rec} + \mathcal{L}_{KL} \}$
11:     $\Phi_{1,\text{dec}} \leftarrow \Phi_{1,\text{dec}} + \alpha_{1,\text{dec}} \nabla_{\Phi_1} \{ \mathcal{L}_{rec} + \mathcal{L}_{KL} \}$
12:    **end for**
13:    # Train discriminator $D_1$
14:    **for** $j < n_D$ **do**
15:      $z_1 \leftarrow \mathcal{N}(0,1), z_2 \leftarrow \mathcal{N}(0,1)$
16:      $y_1 \leftarrow G_1(z_1, c), y_2 \leftarrow G_1(z_2, c)$
17:     $\Phi_{1,\text{disc}} \leftarrow \Phi_{1,\text{disc}} + \alpha_{1,\text{disc}} \nabla_{\Phi_1} \{ \mathcal{L}_{disc} \}$
18:    **end for**
19:    # Train generator $G_1$
20:    **for** $j < n_G$ **do**
21:      $z_1 \leftarrow \mathcal{N}(0,1), z_2 \leftarrow \mathcal{N}(0,1)$
22:      $y_1 \leftarrow G_1(z_1, c), y_2 \leftarrow G_1(z_2, c)$
23:     $\Phi_{1,\text{gen}} \leftarrow \Phi_{1,\text{gen}} + \alpha_{1,\text{gen}} \nabla_{\Phi_1} \{ \mathcal{L}_{gen} + \mathcal{L}_{ms} \}$
24:    **end for**
25: **end for**
26: $z \leftarrow \mathcal{N}(0,1), c \leftarrow \mathcal{N}(0,1)$
27: # Generate fake image $y$
28: $y \leftarrow G_1(z, c)$

---

square are input images from unseen classes in each dataset. The images highlighted in yellow are output images generated by Meta-AVAE. The images in last three row highlighted in green are the samples generated by Meta-FAVAE using the original input images and generated results as the input for Meta-AVAE.

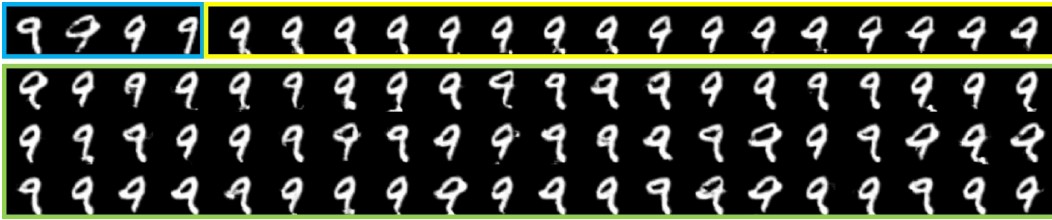

Figure 4: The input images from unseen class in MNIST dataset and generated images from Meta-FAVAE.

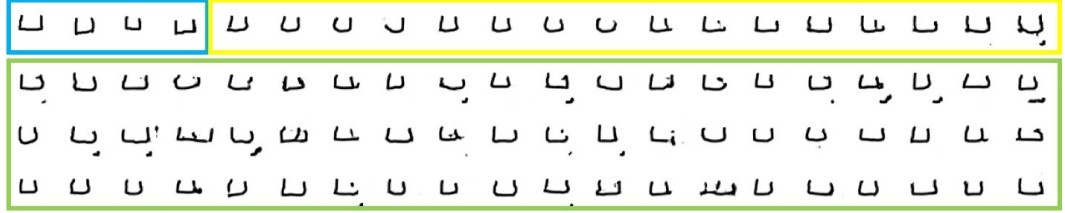

Figure 5: The input images from unseen class in Omniglot dataset and generated images from Meta-FAVAE.

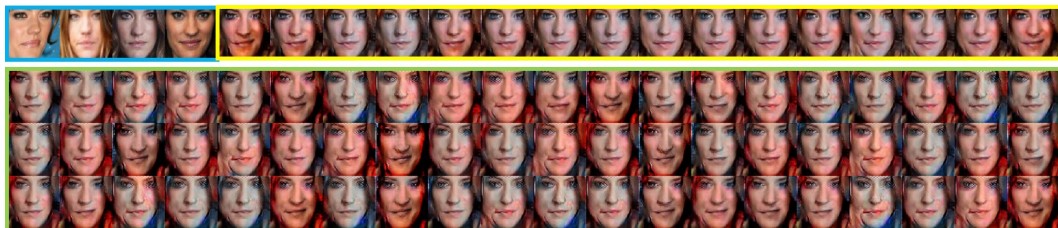

Figure 6: The input images from unseen class in VGG-Face dataset and generated images from Meta-FAVAE.

