# OpenReview forum: "Meta-FAVAE: Toward Fast and Diverse Few-shot Image Generation via Meta-Learning and Feedback Augmented Adversarial VAE"
_ICLR.cc/2022/Workshop/DGM4HSD — ICLR 2022 DGM4HSD workshop Poster_

### Official Review · Reviewer_hG6u · 2022-03-15
**Well-written paper and promising results**

**Rating:** 7
**Confidence:** 3

**Review:**

## Summary

This paper presents Meta-FAVAE, a meta-learning-based adversarial VAE model that uses generated interpolated samples to augment few-shot image generation. Experiments on three few-shot image datasets highlight the better quality of generated images from unseen classes.

## Strengths

S1: This paper is well-written and nicely presented. I enjoy reading it!

S2: Novel idea and promising results.

S3: The related work and background section is very comprehensive.

## Minor weaknesses

M1: How is "image diversity" defined (introduction)? Does it refer to generating different images from similar vectors in the latent space, or images from different domains?

M2: Based on the introduction, it seems the speed of Meta-FAVAE is one major contribution comparing to other related works. However, Section 3 only mentions that it has a fewer number of parameters. I think it would be helpful to also report training time in Section 3.

## Recommendations

R1: Figure 1 looks very good! A minor suggestion is to change the styles of inner loop and outer loop boxes. These two boxes introduce too many intersections with existing lines and make the figure busy. I guess using different translucent background color could help?

---

### Official Review · Reviewer_27rS · 2022-03-28
**Interesting idea, but the results are not very impressive**

**Rating:** 5
**Confidence:** 4

**Review:**

**Summary**

This paper proposes a data-efficient framework for few-shot image generation. The paper is well-motivated, however, it has a few limitations.


**Limitation**
1.  As mentioned in section 2, the proposed method is trained on interpolated samples. If the baseline approaches are not trained on interpolated samples, it would be good to provide results of 1) the proposed method trained without interpolated samples and 2) the baselines trained with interpolated samples since training with interpolated samples can be directly applied to other approaches (and might not be a novel contribution).

2. The results in Fig2 are not very impressive. For instance, the results for 3000 episodes look like an average of the unseen class input. When the input classes have more different images (like the case of 0, 1000, 2000 episodes) the model does not seem to be working well. It would be good to present successful results with more diverse unseen input classes. It would also be useful to present the results of the average and interpolation of input image classes in table 1 and Fig2. It could be possible that these simple settings without model training can easily beat the baselines considered in Fig2.

3. The argument "Due to the limited generation capacity of VAE, this method can only generate 28x28 binary digits" in the introduction could be tenuous. Can it generate other more simple data?

4. (minor) The caption of Fig1 can be improved to be self-contained. For instance, G2, D2 are not referred to in the caption.

---

### Decision · Program_Chairs · 2022-03-26

Accept (Poster)